# Effective removal of mercury from aqueous streams via electrochemical alloy formation on platinum

Cristian Tunsu[1] & Björn Wickman [2]

Retrieval of mercury from aqueous streams has significant environmental and societal importance due to its very high toxicity and mobility. We present here a method to retrieve mercury from aqueous feeds via electrochemical alloy formation on thin platinum films. This application is a green and effective alternative to traditional chemical decontamination techniques. Under applied potential, mercury ions in solution form a stable $PtHg_4$ alloy with platinum on the cathode. A 100 nanometres platinum film was fully converted to a 750 nanometres thick layer of $PtHg_4$. The overall removal capacity is very high, > 88 g mercury per $cm^3$. The electrodes can easily be regenerated after use. Efficient and selective decontamination is possible in a wide pH range, allowing processing of industrial, municipal, and natural waters. The method is suited for both high and low concentrations of mercury and can reduce mercury levels far below the limits allowed in drinking water.

[1] Department of Chemistry and Chemical Engineering, Nuclear Chemistry and Industrial Materials Recycling, Chalmers University of Technology, 41296 Göteborg, Sweden. [2] Department of Physics, Chemical Physics, Chalmers University of Technology, 41296 Göteborg, Sweden. Correspondence and requests for materials should be addressed to B.W. (email: bjorn.wickman@chalmers.se)

Emissions of toxic heavy metals have followed the industrial revolution, and are of significant concern[1,2]. Mercury emissions are particularly worrisome due to mercury's high toxicity, and high rates of spreading and accumulation in natural waters[3−5]. Estimations show that mercury is a serious toxic threat, affecting tens of millions of people globally[6,7]. Anthropogenic sources of mercury include gold mining, burning of fossil fuels, metal production, cement production, waste incineration, contaminated sites, chloralkali industries, and dental amalgam production and use[8]. The global atmospheric emissions of mercury have been estimated to be in the range of 1010—4070 tonnes per year[5], of which anthropogenic sources account for 30%. The rest is attributed to re-emission of mercury from oceans and lakes (60%), and to natural sources, e.g. volcanic and geothermal activities (10%). It was estimated that 1130 Gg mercury was released from anthropogenic sources to the environment between the years 1850 and 2010, of which 336 Gg were emitted directly to the atmosphere[9]. A significant part of this affects water ecosystems[10]. Constant efforts have, and are being carried out to mitigate emissions, e.g. employing cleaner or alternative processes, trapping harmful substances to prevent their release, chemical conversion to more stable or less toxic species etc. Despite these efforts, heavy metal pollution remains a serious problem worldwide[7].

Mercury has an overall high mobility, which facilitates its environmental cycling and uptake by living organisms[5,8]. The mobility and spread of mercury are closely connected to water and movement of natural waters[11]. As water is essential for life, and has significant contribution to the cycling of mercury in the environment, its contamination is a key issue. Current solutions to reduce mercury levels in aqueous streams include precipitation, flocculation, absorption, ion exchange, and solvent extraction[12]. Precipitation, e.g. with sulphur-based reagents or via reaction with selenium, requires addition of chemicals to facilitate decontamination, and physical separation of the precipitate formed is required to completely remove mercury from the stream[13]. This method poses limitations for large volumes containing trace amounts of mercury. In addition, since metal sulphates tend to have low solubility in general, undesired co-precipitation of other metal ions can be problematic for solutions with complex chemical composition. The selectivity of absorption and ion exchange techniques is also limited by chemical complexity. Very low or very high metal concentrations, and small or large feed volumes are other factors that limit use. Furthermore, it can be both difficult and costly to dispose of, or regenerate resulting contaminated absorption materials. For these reasons, the development of improved technologies to remove toxic heavy metals from aqueous streams is of high importance, specifically chemical-free processes that do not generate secondary wastes, and are also highly effective at very low metal concentrations.

In recent years, other approaches to remove mercury from aqueous streams have been suggested and evaluated[14−18]. Among them is the incorporation of mercury ions in a solid and stable metallic alloy, which is afterwards removed from solution. For this, soluble mercury ions ($Hg^{2+}$, $Hg_2^{2+}$, $CH_3Hg^+$) are reduced to elemental mercury ($Hg^0$), followed by subsequent amalgamation with a metal. Several such systems have been described[19−24]. One example is the use of gold nanoparticles coated with sodium citrate, where the latter acts as electron donor to facilitate reduction of mercury and formation of $Au_3Hg$[22]. Smooth gold surfaces were found to be selective towards $Hg^0$, and nano-structured gold surfaces showed affinity for different mercury species ($Hg^0$, $Hg^{2+}$, $CH_3Hg^+$)[24]. While $Hg^0$ absorption occurred via amalgamation, $Hg^{2+}$ and $CH_3Hg^+$ adsorption was ascribed to the catalytic activity of the nano-structured gold surface. Brass shavings have shown potential for decontamination through a process where zinc oxidizes and donates its electrons to reduce mercury ions, which subsequently form an alloy with copper[23]. Sole copper or tin have also been used, and here the metals act as both electron donors and alloying components[19,21]. However, these methods have prominent drawbacks, notably stability, as the oxidized component will, under most conditions leach from the material and contaminate the water stream. This was reported in the aforementioned studies. The chemical stability of, e.g. brass, copper, and tin at low pH is an issue which further complicates decontamination of acidic wastes. Moreover, additional physical separation, e.g. sedimentation, is typically required to isolate the formed alloy[22]. In another study, the use of metallic mossy filters eventually caused blocking of the filters with sludge (mainly tin hydroxide), and a collapse of the system[19]. Electro-chemistry is another technique that has potential for retrieval of metal ions. Mercury is noble enough to have its reduction potential inside the water stability window, making it possible to electroplate metallic mercury from aqueous solutions[25]. However, direct electroplating is not practical for decontamination, as metallic mercury is a liquid at ambient conditions, and its vapour pressure and concentration in air increases significantly with increasing temperature, making the system less stable[26].

Nonetheless, it is clear that alloy formation (amalgamation) and electrochemistry are interesting routes to develop improved methods for removal of mercury. Amalgamation requires an interaction between metals, which means that mercury ions in solution need to be reduced to $Hg^0$. By doing this electro-chemically, under controlled potential the use of chemical reducing agents is avoided, a clear advantage. A particularly promising way to carry out this can be to use the platinum–mercury system. The advantages include high stability in solution and high theoretical saturation capacity. A platinum atom can bind up to four mercury atoms, as $PtHg_4$, while a gold one binds twelve times less mercury in its most stable form, $Au_3Hg$[22,27]. The solubility of liquid mercury in platinum, and of platinum in liquid mercury are low at ambient conditions, e.g. only $5 \times 10^{-4}$% platinum atoms are soluble in liquid mercury at 25 °C[27], which makes it highly unpractical for decontamination. However, by applying a negative potential to the platinum, it is possible to increase the saturation solubility immensely. In fact, it is well known that electrochemistry can be used to form thin layers of platinum–mercury alloys[27]. Three intermetallic compounds can form, $PtHg$, $PtHg_2$, and $PtHg_4$, the last being distinctly favoured thermodynamically (enthalpies of formation: $-9.2 \pm 0.6$, $-14.8 \pm 0.4$, and $-16.4 \pm 0.4$ kJ $mol^{-1}$, respectively). Platinum–mercury alloys have high stability, and temperatures up to 900 °C are needed for their complete decomposition[28,29]. The $PtHg_4$ amalgam exhibits sufficient adhesion to the platinum substrate, enabling cleaning of reacted platinum surfaces with cold nitric acid, alcohol, and freon jet without any amalgam losses[30].

Electrochemical formation of platinum–mercury alloys has been documented[28,29,31] but these efforts focused on studying the solid-state interactions occurring at the metallic interface. While electrochemical alloy formation between platinum and mercury ions in solution appears plausible for decontamination purposes, this prospect has not been explored. To our knowledge, potential-controlled retrieval of ionic mercury as platinum alloy from solutions containing environmentally relevant amounts of mercury (mg $L^{-1}$ or lower) has not been thoroughly investigated. Instead, the previous literature has focused on fast electro-chemical amalgamation from very concentrated solutions (typically, 300 s; 12 g $L^{-1}$ monovalent mercury) for characterization of the compounds formed using X-ray Diffraction (XRD), Scanning Electron Microscopy/Energy Dispersive Spectroscopy (SEM/EDS), Thermal Gravimetric Analysis (TGA), and X-ray Photo-electron Spectroscopy (XPS). The influence of parameters

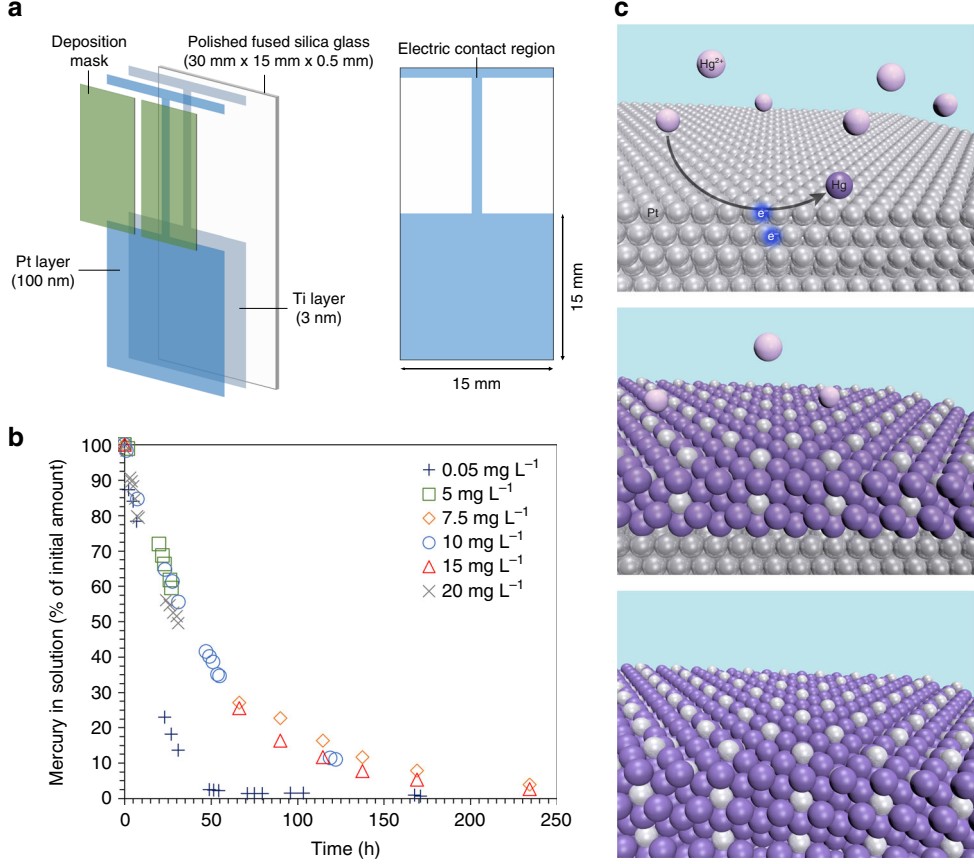

**Fig. 1** Electrochemical alloy formation between divalent mercury in solution and metallic platinum. **a** Schematic representation of the platinum nano-film electrodes used. **b** The influence of initial mercury concentration in solution on the decontamination efficiency. The electrolytes were 50 mL 1 mol L$^{-1}$ nitric acid solution with 0.05—20 mg L$^{-1}$ mercury, as follows: 0.05 mg L$^{-1}$ (dark-blue pluses), 5 mg L$^{-1}$ (green squares), 7.5 mg L$^{-1}$ (orange diamonds), 10 mg L$^{-1}$ (blue circles), 15 mg L$^{-1}$ (red triangles), and 20 mg L$^{-1}$ (grey crosses). Working electrode: 100 nm platinum film (2.25 cm$^2$ area). Counter electrode: platinum wire. Reference electrode: Hg/Hg$_2$SO$_4$. Potential = 0.16 V vs. RHE. **c** Schematic representation of the alloying process. Divalent mercury ions in solution (light purple colour) are first reduced on the platinum surface (silver colour) to elemental mercury. Elemental mercury (darker purple) forms thermodynamically stable PtHg$_4$ with the platinum atoms. Upon formation of the first layers of PtHg$_4$, mercury atoms penetrate the metallic alloy film to grow the alloy

important for decontamination (pH, uptake time, mercury concentration in solution, presence of other ionic species, electrochemical regeneration and re-use of the platinum substrate, uptake from solutions containing trace levels of mercury, and complete saturation of the substrate) has not been reported. Stability is a key issue for using this type of system for mercury retrieval, especially at low pH and long exposure times. Given these, the method of using platinum to form electrochemical alloys with mercury ions from solution for decontamination purposes appears plausible. However, to function as a practical method for decontamination, it must be possible to form relatively thick and uniform layers of PtHg$_4$, preferably tens or hundreds of nm. In addition, to be truly attractive, the process must be effective at lowering mercury in solution below the limits set by the World Health Organization for drinking water (6 µg L$^{-1}$ inorganic mercury)[32].

In this study, we address the above-mentioned aspects. We report on the decontamination of divalent mercury ions in solution via electrochemical alloy formation on thin platinum layers (100 nm). We analyse the alloying process in detail, in order to evaluate the limitations, and the prospects for large scale decontamination applications. We show that alloy films with a thickness of 750 nm can be formed from a 100 nm platinum film, which is in perfect agreement with the 760 nm theoretical thickness expected for complete conversion of 100 nm platinum

to PtHg$_4$. The alloying process is not affected by pH, and it is possible to remove mercury well below the limits allowed in drinking water. The system is efficient also in the presence of other cations and anions in solution, and reversible. By increasing the surface area using platinum nanoparticles, the time required for retrieval decreases significantly; a decontamination efficiency above 99.4% was achieved for a solution containing 10 mg L$^{-1}$ mercury. This opens up new possibilities for cleaner and more efficient methods to remove mercury in a large number of applications, from treatment of highly acidic industrial streams to treatment of natural waters.

## Results and Discussion

**Alloy formation and the influence of mercury concentration.** Deposition masks were used in the preparation of the electrodes, to obtain the pattern schematized in Fig. 1a. This design allowed us to control and estimate with precision the number of platinum atoms in contact with the solution (active during retrieval). The contaminated feeds contained divalent mercury nitrate dissolved in nitric acid solutions. Unused working electrodes, and working electrodes previously used to retrieve mercury (loaded at 25% of the PtHg$_4$ stoichiometric saturation limit) were each immersed for 30 h in 1 mol L$^{-1}$ nitric acid solution containing 10 mg L$^{-1}$ mercury. In both cases, there were no decreases or increases of the aqueous platinum or mercury concentrations. This showed

that the platinum and the platinum–mercury alloy layers are stable at low pH in the absence of applied electrical potential. Based on the cyclic voltammograms in pure nitric acid, and in nitric acid containing mercury, $-0.5$ V vs. Hg/Hg$_2$SO$_4$ (0.16 V vs. the reversible hydrogen electrode, RHE) was selected for electrochemical retrieval (see Supplementary Figure 1).

Mercury concentration in solution plays an important role in practical decontamination applications; retrieval needs to be effective at low and high levels of mercury. Figure 1b shows the retrieval from solutions with an initial mercury content between 0.05 and 20 mg L$^{-1}$. The data was normalized to the initial mercury concentration in solution, to allow easier comparison. Between 30 and 40 h were needed for the mercury concentration in a solution containing 10 mg L$^{-1}$ mercury to drop below 50%. After 130 h, less than 10% of the initial mercury was present in solution. Retrieval is faster in the beginning, and it slows down with passing time. We interpret this by looking at the alloy formation mechanism as a multi-step process. The formation of PtHg$_4$ from metallic platinum and ionic mercury proceeds via a sequential route of electrochemical reduction of mercury, followed by formation of the alloy. Potential routes for these are shown in Eqs 1–10.

$$Hg^{2+} + 2e^- \rightarrow Hg^0 \tag{1}$$

$$2Hg^{2+} + 2e^- \rightarrow Hg_2^{2+} \tag{2}$$

$$Hg_2^{2+} + 2e^- \rightarrow 2Hg^0 \tag{3}$$

$$Pt + Hg \rightarrow PtHg \tag{4}$$

$$Pt + 2Hg \rightarrow PtHg_2 \tag{5}$$

$$PtHg + Hg \rightarrow PtHg_2 \tag{6}$$

$$PtHg_2 + 2Hg \rightarrow PtHg_4 \tag{7}$$

$$Pt + 4Hg \rightarrow PtHg_4 \tag{8}$$

$$Pt + 4Hg^{2+} + 8e^- \rightarrow PtHg_4 \tag{9}$$

$$Pt + 2Hg_2^{2+} + 4e^- \rightarrow PtHg_4 \tag{10}$$

First, divalent mercury in solution is reduced on the platinum surface (Eq. 1). Mercury atoms will then move to subsurface positions by a place exchange mechanism with platinum atoms, followed by penetration into bulk platinum. The latter involves the inward shift of mercury atoms to attain the maximum coordination number with platinum[33]. This creates holes in bulk platinum, which allows further diffusion of mercury atoms. Diffusion is facilitated by the chemical potential gradient of mercury built up between the mercury deposit and the bulk platinum. According to the aforementioned study[33], the stoichiometry of the subsurface PtHg alloy changes from PtHg$_2$, when a second monolayer of mercury is deposited (Eq. 6), to the favoured PtHg$_4$ when additional monolayers of mercury are deposited (Eq. 7). Thus, PtHg$_4$ species are preferably formed over PtHg and PtHg$_2$ if sufficient bulk mercury is present, and if the reaction time is appropriate. The overall process is described by Eq. 9. PtHg$_4$ is stable thermodynamically, having a negative enthalpy of

formation, and this will stabilise mercury and prevent its dissolution[34]. This negative formation energy, together with the applied potential, provides the driving force to maximize coordination of mercury and platinum. After the first layers of alloy is formed, additional mercury atoms need to penetrate into the metallic alloy film to grow the alloy (Fig. 1c). It was reported that mercury is more abundant in the first layers after deposition[34], which indicates that the diffusion of mercury is a slower process, and most likely the rate determining step in the decontamination. The reduction of available active platinum atoms on the surface slows down the absorption of more mercury. These observations seem to correlate with those made by Wang et al.[35], and Ojea-Jiménez.[22], who also reported slow inward metal inter-diffusion of mercury, albeit on gold particles. Past research states that amalgamation is unlikely to occur by migration of platinum atoms through the reaction product following dissociation of the atoms from the platinum lattice, and rather by transport of mercury atoms through the amalgam layer[30]. We have performed tests at higher temperature, and found out that retrieval is significantly faster upon increasing the temperature. This agrees very well with the diffusion of mercury ions being the rate limiting step (see Supplementary Figure 4).

The fact that the intermetallic compounds formed at the mercury–platinum interface still allow for further reactions between the surface mercury and the bulk platinum is of high importance. We believe this property is vital for the decontamination of concentrated streams, as formation of relatively thick alloy layers at the interface will not completely stop further platinum–mercury interactions but rather slow them down. In this study, we focused on small platinum surfaces (flat 2.25 cm$^2$ films), which are sufficient to get an understanding of the electrochemical process at laboratory scale. For practical applications, the slow diffusion of mercury in the alloy can be mitigated by using electrodes with sufficiently large surface area in relation to the amount of mercury in solution. This hypothesis has been validated here by using larger surface area electrodes, and will be presented below. For industrial use, the electrodes can be designed to have large active surface by employing, e.g. packed bed columns, porous construction, or nanoparticles.

At significantly lower mercury concentrations of 0.05 mg L$^{-1}$, the decontamination process was much faster. Over 75% of the mercury was retrieved in one day (Fig. 1b), and the efficiency was over 99% after 171 h. This corresponds to 0.35 µg L$^{-1}$ mercury left in solution, which is well below the acceptable limit in drinking water[14,32]. We correlate this with the aforementioned slower inward metal inter-diffusion of mercury once several layers of alloy are formed. These effects should be less preeminent at low concentrations due to increased number of interactions per available platinum active surface area. For a solution containing 10 mg L$^{-1}$ mercury, assuming complete retrieval of mercury as PtHg$_4$, about 25% of the saturation capacity of the working electrode is achieved.

Figure 2 shows the XRD patterns of a working electrode before and after electrochemical treatment. The predominant phase formed was PtHg$_4$, as expected from the thermodynamics[27]. The PtHg$_4$ pattern was registered on a 100 nm platinum electrode after electrochemical treatment, where the platinum film was loaded at about 22.5% of the PtHg$_4$ stoichiometric saturation limit during 122 h. While it is possible to form other alloy phases under electrochemical conditions, e.g. PtHg$_2$, we did not see any clear signs of phases other than PtHg$_4$. This is likely to be explained by the fact that the experiments were carried out for long times. As PtHg$_4$ is the alloy phase with the lowest energy, it is expected that this dominates if the system is allowed time to relax, and relatively thick films are formed.

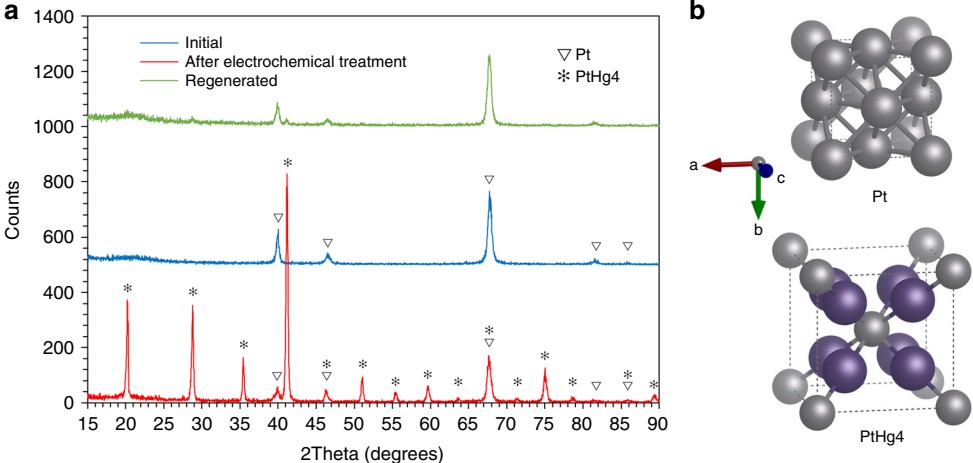

**Fig. 2** X-ray Diffraction characterization of electrodes before and after electrochemical treatment, and after regeneration. **a** The XRD patterns of 100 nm platinum electrodes before (blue line) and after electrochemical treatment (red line), and after regeneration (green line). The number of counts on the x-axis has been offset for clarity (plus 500 and 1000 counts for the initial and regenerated electrodes, respectively). **b** Schematic illustrations of the unit cells for platinum and $PtHg_4$

**Saturation of 100 nm platinum electrodes**. Saturation of a 2.25 $cm^2$ 100 nm platinum film was done in 50 mL solution containing 75 mg $L^{-1}$ mercury, well above the theoretical $PtHg_4$ saturation limit. After 48 h the electrode was taken out of solution, and analysed with SEM/EDS. The EDS analysis suggested a Pt:Hg weight ratio of 0.22. Calculations based on the ICP-MS analysis of the resulting electrolyte suggested a ratio of 0.2. This is slightly less than the expected value of 0.24 and is likely a result of excess mercury on the surface of the electrode. Measurements in the SEM image from a cross section of the formed alloy showed a thickness of about 750 nm, in good agreement with the 760 nm value expected for full transformation of a 100 nm platinum film to $PtHg_4$. This is important since a practical system needs to have good loading capacity, and the results show that 100 nm platinum can easily be saturated to $PtHg_4$. However, the saturated film had less adhesion to the glass substrate, which caused the film to crack and eventually peel off in certain spots (see Supplementary Figure 3). Such behaviour was not seen for films loaded below the saturation limit.

**pH dependency of the alloying process**. The reactions described by Eqs. 1–10 are not dependent on the concentrations of protons or hydroxyl ions in solution, thus they are not dependent on the pH of the mercury-containing solution. The platinum surface, however, will be different at a certain potential relative to the standard hydrogen electrode (SHE) at different pH[36] (see Supplementary Discussion, pH dependency). To avoid influences from changes on the platinum surface, we chose to study the pH dependency of the alloy formation at a fixed potential, 0.16 V vs. RHE. The RHE scale relates to SHE according to: $E_{RHE} = E_{SHE} + 0.059$ pH.

Figure 3 shows the results of several mercury retrieval experiments from solutions with pH in the range 0–6.6. In all experiments, the electrolyte was nitric acid solution with an initial mercury concentration of 10 mg $L^{-1}$. The ionic strength was held constant by balancing the amount of acid with sodium nitrate to yield 1 mol $L^{-1}$ nitrate. There was no significant pH dependency for the formation of the alloy in the pH range examined. Decontamination was as effective at very low pH as it was close to neutral pH, a key advantage for practical applications.

**Interferences during alloy formation**. Selectivity is desired for practical applications, in the sense that alloy formation is not

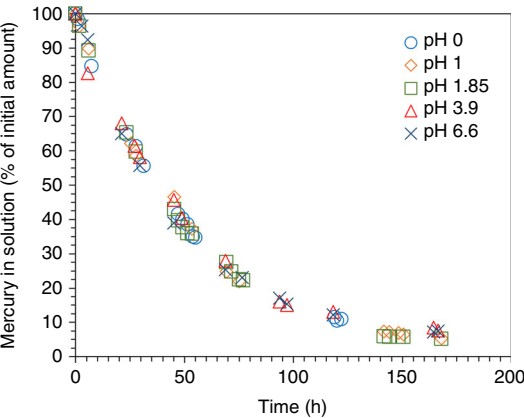

**Fig. 3** Influence of pH on the decontamination process. The plot shows the decrease of mercury concentration from solutions containing 10 mg $L^{-1}$ mercury and various amounts of nitric acid: pH 0 (blue circles), pH 1 (orange diamonds), pH 1.85 (green squares), pH 3.9 (red triangles), and pH 6.6 (dark-blue crosses). $[NO_3^{-}] = 1$ mol $L^{-1}$. Working electrode: 100 nm platinum film (2.25 $cm^2$ area). Counter electrode: platinum wire. Reference electrode: $Hg/Hg_2SO_4$. Potential = 0.16 V vs. RHE

hindered by the presence of other species in solution, and preferably that retrieval of other species does not occur together with retrieval of mercury. Selectivity was studied using 1 mol $L^{-1}$ nitric acid solution containing, apart from 10 mg $L^{-1}$ mercury, 10 mg $L^{-1}$ each of calcium, cadmium, copper, magnesium, manganese, sodium, nickel, lead and zinc, and 20 mg $L^{-1}$ iron. During the experiment, mercury in solution decreased similar to the results in Figs. 1 and 3, to about 7% of its initial amount after 168 h. The concentrations of calcium, cadmium, iron, magnesium, sodium, nickel, and zinc in solution remained constant. The amounts of copper, manganese, and lead decreased by about 37%, 10%, and 72%, respectively (Fig. 4a). SEM/EDS analysis revealed the presence of copper on the working electrode (the cathode), and it is interesting to note that the addition of copper did not affect the uptake of mercury. Manganese and lead were found on the platinum counter electrode (the anode), which is explained by the fact that the half-reactions for formation of $MnO_2$ and $PbO_2$ can be favoured over water oxidation on a platinum electrode[25].

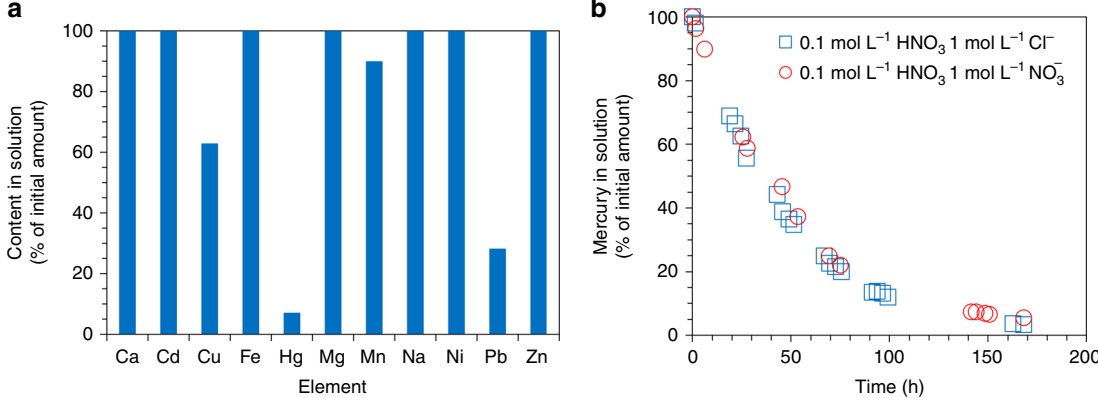

**Fig. 4** Selectivity during alloy formation. **a** Element selectivity during retrieval for 168 h from 50 mL 1 mol L$^{-1}$ nitric acid solution containing 10 mg L$^{-1}$ of each calcium, cadmium, copper, mercury, magnesium, manganese, sodium, nickel, lead and zinc, and 20 mg L$^{-1}$ iron. Working electrode: 100 nm platinum film (2.25 cm$^2$ area). Counter electrode: platinum wire. Reference electrode: Hg/Hg$_2$SO$_4$. Potential = 0.16 V vs. RHE. **b** The decrease of mercury concentration upon electrochemical treatment of (i) 50 mL solution containing 0.1 mol L$^{-1}$ nitric acid, 1 mol L$^{-1}$ sodium chloride, 10 mg L$^{-1}$ divalent mercury (blue squares), and (ii) 50 mL solution containing 0.1 mol L$^{-1}$ nitric acid, 1 mol L$^{-1}$ [NO$_3^-$] as sodium nitrate, and 10 mg L$^{-1}$ mercury (red circles). Working electrode: 100 nm platinum film (2.25 cm$^2$ area). Counter electrode: platinum wire. Reference electrode: Hg/Hg$_2$SO$_4$. Potential = 0.16 V vs. RHE

Chloride anions in solution can affect the speciation of mercury due to formation of chloroanions (Eqs. 11 and 12).

$$Hg^{2+} + 3Cl^- \leftrightarrow [HgCl_3]^- \tag{11}$$

$$Hg^{2+} + 4Cl^- \leftrightarrow [HgCl_4]^{2-} \tag{12}$$

To investigate if chloroanions would interfere with the alloying process, electrochemical retrieval was studied from a solution containing 10 mg L$^{-1}$ divalent mercury and 1 mol L$^{-1}$ sodium chloride in 0.1 mol L$^{-1}$ nitric acid. Figure 4b shows a comparison between the data obtained for this test, and the data corresponding to the retrieval from a solution containing the same amount of mercury in 0.1 mol L$^{-1}$ nitric acid with 1 mol L$^{-1}$ nitrate ions (under the same experimental conditions). The uptake of mercury showed similar behaviour, suggesting the dissociation equilibrium in reactions 11 and 12 is faster than the limiting step for alloy formation, i.e. the diffusion of mercury in PtHg$_4$.

Natural waters typically contain organic matter which facilitates formation of organic mercury species, e.g. methylmercury (CH$_3$Hg$^+$). How the presence of such species in solution affects the alloying process was not studied here. Mercury in aquatic environments cycles between various chemical species, including photodecomposition of methylmercury to inorganic mercury[37,38]. Decomposition of methylmercurychloride to divalent mercury can take place immediately under UV exposure, depending on the intensity of the radiation[37]. An in-situ study of lake water showed that the annual rates of methylmercury photodegradation in surface waters are almost double the estimated external inputs of methylmercury from rain, snow, streamflow, and land runoff[38]. For these reasons, we do not completely rule out the applicability of the method on contaminated solutions containing organic matter. Mercury levels in such streams can still be decreased under UV irradiation, as the ionic inorganic species from the photodegradation of organic mercury will form alloys with platinum. If the organic species can easily move to the surface of the working electrode (e.g. attraction of the positive methylmercury ion), and the energetics favour decomposition, alloy formation should also occur in the absence of UV radiation.

**Regeneration of used electrodes**. The chronoamperometry data registered upon retrieval of mercury from 50 mL 1 mol L$^{-1}$ nitric acid solution containing 10 mg L$^{-1}$ mercury (presented in Figs. 1 and 3 above) showed a reduction current in the range of 40 μA (see Supplementary Figure 2). This was used as basis to study the electrochemical regeneration of electrodes previously used to retrieve mercury. An electrode loaded to about 12% of the PtHg$_4$ saturation limit was immersed in pure 1 mol L$^{-1}$ nitric acid solution, and a 40 μA oxidation current was applied. Figure 5a shows the increase of mercury content in the nitric acid solution over time, and the corresponding potential. Regeneration was significantly faster than retrieval, and mercury was released from the electrode with very high efficiency (>95% in 10 h.). After regeneration, the electrode was analysed with XRD. Figure 2 shows the XRD pattern, which reveals that platinum is, again, the dominating phase. The regenerated film remained attached to the glass substrate. A regenerated electrode was successfully re-used for another retrieval–stripping cycle. The efficiency was similar to previous observations, the bulk mercury being retrieved in about seven days, followed by much faster release in 1 mol L$^{-1}$ nitric acid when applying a 40 μA oxidation current. The platinum layer was stable during regeneration and re-use.

**Use of large surface area platinum electrodes**. The retrieval process was investigated with electrodes onto which ca. 0.02 g of 50%wt. platinum nanoparticles on carbon black were deposited. The active platinum surface area in this electrode could theoretically be more than 1000 times higher than that of the flat films on glass used above (the 50%wt Pt on carbon powder has a platinum surface area of 110 m$^2$ g$^{-1}$). With the rather high loading, and the fact that Nafion binder was used to fixate the catalyst powder on the electrode, it is likely that a significantly smaller effective area was available for mercury absorption. However, this electrode should still have a significantly larger area than the flat platinum films above. Figure 5b shows the result for such a test. Retrieval was about 20 times faster than for the flat films, which confirmed our hypothesis that a large enough surface area will mitigate the downsides related to slow inward metal diffusion after several layers of platinum–mercury alloy are formed. Over 99.4% of the mercury was retrieved in 24 h from a 1 mol L$^{-1}$ nitric acid solution containing 10 mg L$^{-1}$ divalent mercury ions.

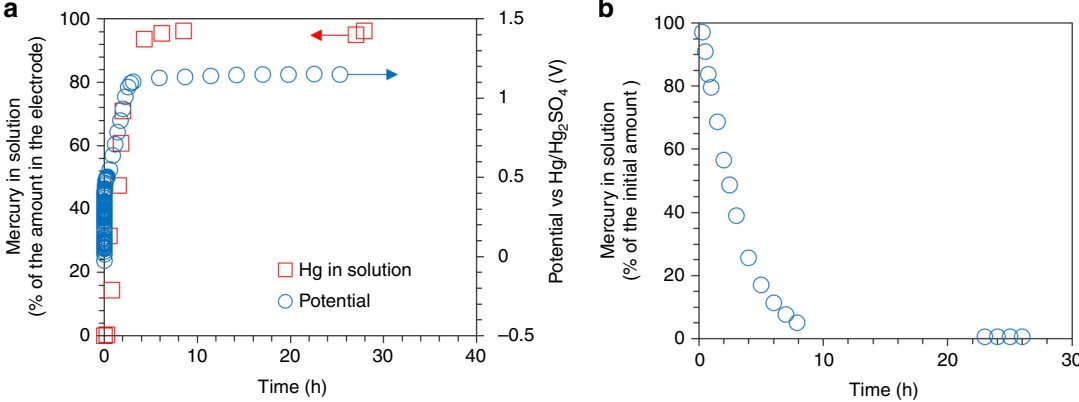

**Fig. 5** Regeneration of nano-film electrodes and the use of large surface area electrodes. **a** The increase of mercury concentration in pure 1 mol L$^{-1}$ nitric acid solution upon applying a 40 μA current to a 100 nm platinum electrode previously used for mercury retrieval (red squares, left axis), and the recorded potential (blue circles, right axis). Counter electrode: platinum wire. Reference electrode: Hg/Hg$_2$SO$_4$. **b** Retrieval of mercury from 50 mL 1 mol L$^{-1}$ nitric acid solution containing 10 mg L$^{-1}$ mercury. Working electrode: 50%wt. platinum nanoparticles on carbon black, deposited on carbon paper support. Counter: platinum wire. Reference: Hg/Hg$_2$SO$_4$. Working potential = 0.16 V vs. RHE

**Advantages for the proposed method.** The method described here has notable advantages over precipitation, ion exchange, and solvent extraction decontamination processes for mercury. The method does not rely on the addition of chemicals to contaminated solutions, and it does not require, e.g. organic extractants or specifically tailored resins. This eliminates the subsequent separation of any insoluble compounds, which is mandatory to isolate the precipitated toxic metals from solution.

Conventional treatment of large volumes of contaminated solution have additional energy demands, e.g. pumping through absorbents, mixing, filtration etc. The proposed technology was shown to have low energy requirements for both retrieval and regeneration. In theory, portable systems can be designed to be used on site, and they can be powered by batteries and solar cells. Mercury is recovered at the cathode, in a very stable form, avoiding additional processing of the feed, e.g. filtration and sedimentation. Electrochemical regeneration of the working electrodes for re-use is effective, and does not generate any other streams aside from the solution in which mercury is released. This can be a relatively small volume, and the contained mercury can be further re-used into suitable applications. A key advantage is the efficiency of the process in a wide pH range. This further strengthens the potential to find practical use to retrieve mercury from acidic industrial wastes, as well as from waters contaminated with inorganic mercury species. The former is important due to the aforementioned limitations of thiol resins to process oxidizing feeds, e.g. nitric acid solutions containing mercury. The latter is crucial given the significant role water plays in the cycling of mercury in the environment, and its role in sustaining life on the planet. It was shown that the system is efficient at both high and low initial concentrations of mercury in solution. This makes treatment of streams with very low mercury content possible, which is usually more challenging for precipitation.

The formation of PtHg$_4$ was not hindered by the presence of calcium, cadmium, copper, iron, magnesium, manganese, sodium, nickel, lead, zinc and, chloride ions, a significant advantage for treating chemically-complex streams. The concentrations of copper, manganese and lead in solution decreased concomitant with mercury. This is not problematic for decontamination, as the presence of some of these heavy metals in waters is not desired. Copper and lead also bind to thiol-based ligands for mercury removal[39].

Platinum is a material with a high cost, which raises concerns about the practical applicability of this method. It was shown here

that efficient retrieval can be achieved on thin films and nanoparticles. One particular area of applicability is retrieval of mercury from natural waters (in which mercury is preferably present as inorganic species), which carries significant importance. These feeds generally contain low levels of mercury, but due to the large flows, the total amounts are still worrisome. Efficient uptake of mercury from such streams will not load the electrodes significantly, providing large enough active area. One platinum atom binds up to four mercury atoms upon formation of PtHg$_4$, making the overall removal capacity very high, more than 88 g mercury per cm$^3$. Since used electrodes can be easily regenerated, they can be re-used. Platinum has high chemical stability, mitigating possible losses due to dissolution or acid attack. This also prevents further spread of unwanted metallic ions in the feed by, e.g. unwanted chemical interactions of the active metal on the electrode with constituents of the feed. Stability under excessive loading or saturation is a concern, e.g. cracks and losses of adhesion. Nonetheless, the elements in such a system can be easily reclaimed by thermally decomposing the alloy into volatile mercury, and platinum, which can be further re-used to prepare new electrodes. We have employed here the simplest design, e.g. nano-film on a flat surface. This was to get an understanding of the alloying process and make a simple evaluation of loading under different conditions and regeneration. We have shown that a design with larger surface area makes retrieval significantly faster. For practical applications, possible designs include porous platinum layers, coated nanoparticles immobilized in a column or mesh, porous structures, and even bulk platinum electrodes.

## Methods

**Preparation of platinum nano-film electrodes.** Polished fused silica glass (Mark Optics Inc.), cut to 15 mm × 30 mm × 0.5 mm, were cleaned under sonication, first with acetone, then isopropanol, and finally with pure water. The duration of each step was 15 minutes. A 3 nm adhesion layer of titanium was deposited on one side of the glass using physical vapour deposition (Lesker PVD 225), followed by deposition of a 100 nm platinum layer using the same technique. Deposition masks were used to obtain the pattern schematized in Fig. 1. A thin (1 mm) connection strip extending to the top of the glass facilitated the electric contact to connect the electrode to the potentiostat. This contact was made on the top side of the glass using polymer-covered copper wire attached with copper conductive tape, and sealed with hot glue. During the experiments, the electrodes were fixed so that the solution level was just above the 15 × 15 mm platinum area, and below the contact region. This was to make sure that no mercury could be absorbed by the wire, the copper tape, or the hot glue. The solution level was within ± 1 mm of the thin platinum connection strip, giving an uncertainty in the estimation of the number of

active platinum atoms of < 1% (given a ± 0.1 nm accuracy in layer thickness and ± 1 mm on the immersion level).

**Preparation of large surface area electrodes**. An electrode with significantly larger surface area was prepared using platinum nanoparticles on carbon catalyst (50%wt. platinum on carbon black, Fuelcellstore). 0.02 g of catalyst powder was sonicated in 500 μL isopropanol and 250 μL 5% Nafion solution (D521, Alfa Aesar), and drop casted onto a 2 cm$^2$ piece of carbon paper (Sigracet GDL 24 BC), then subsequently dried in an oven at 60 °C. The carbon paper was contacted in the same way as the glass above, and the solution level was kept below the glue during the measurements.

**Aqueous feeds**. The electrolyte solutions were prepared by mixing adequate amounts of pure water (MilliQ, Millipore, > 18 MΩ cm$^{-1}$), high purity nitric acid solution (65%, Suprapur, Merck), and metal standard solutions (1001 ± 2 mg L$^{-1}$, Ultra Scientific Analytical Solutions). The mercury standard solution contained mercuric nitrate in 2% nitric acid. Calcium, iron, magnesium, manganese, sodium, nickel, cadmium, copper, lead, and zinc standard solutions were used in the interferences study. These contained the corresponding metal nitrates dissolved in dilute nitric acid. The electrolytes were titrated with sodium hydroxide (0.1 mol L$^{-1}$, Titrisol, Merck) to determine their acidity, and their pH was measured with a pH meter. Where needed, e.g. for the pH dependency investigations, the ionic strength was kept constant by addition of solid sodium nitrate to achieve 1 mol L$^{-1}$ nitrate concentration. In the regeneration experiments, 1 mol L$^{-1}$ nitric acid solution prepared from pure water and stock high purity nitric acid was used as electrolyte.

**Blanks and electrochemical investigations**. Unused 100 nm platinum electrodes, and partly loaded nano-film electrodes from prior retrieval tests were immersed for 30 h in 1 mol L$^{-1}$ nitric acid solution containing 10 mg L$^{-1}$ mercury. This was to see if the mercury levels decrease in the absence of electrical potential, and if the metallic films are stable in acidic media.

For the electrochemical studies a platinum wire was used as counter electrode (anode), and a Hg/Hg$_2$SO$_4$ electrode (SI Analytics) was used as reference. A potentiostat (Gamry, Reference 600) was used to control the potential or the current, and to acquire electrochemical data. Before retrieval, the working electrodes (cathodes) were washed with isopropanol, then with pure water, followed by cyclic voltammetry in 0.5 mol L$^{-1}$ sulfuric acid solution between 0 and 1.4 V vs. RHE. A final washing with pure water was done to assure a clean surface. All investigations, except the retrieval test at 40 °C, were performed at ambient temperature, 20 ± 1 °C ( ± reflects the temperature variation during the runtime of the experiments, monitored daily with a thermometer). The volume of electrolyte used was 50 mL. The potential in the retrieval studies was 0.16 V vs. RHE.

To study the influence of pH on the alloying process, solutions with pH in the range 0–6.6, with 1 mol L$^{-1}$ nitrate and 10 mg L$^{-1}$ mercury, were used. To study the influence of mercury concentration in solution, 1 mol L$^{-1}$ nitric acid solutions containing between 0.05 and 20 mg L$^{-1}$ mercury were used. The metal concentration in solution was monitored by sampling 0.1 mL of the electrolyte before, during and after electrochemical treatment. Saturation of a 100 nm platinum nano-film was investigated using 1 mol L$^{-1}$ nitric acid solution containing 75 mg L$^{-1}$ mercury, almost double the stoichiometric amount needed to convert all the platinum into PtHg$_4$. The electrochemical behaviour of other cations in solution was studied using a 1 mol L$^{-1}$ nitric acid solution containing 10 mg L$^{-1}$ each of calcium, cadmium, copper, mercury, magnesium, manganese, sodium, nickel, lead and zinc, and 20 mg L$^{-1}$ iron. To study the interference of chloride ions, an electrolyte with 10 mg L$^{-1}$ mercury in 0.1 mol L$^{-1}$ nitric acid solution and 1 mol L$^{-1}$ sodium chloride was used. Regeneration of nano-film electrodes used for mercury retrieval was done in 1 mol L$^{-1}$ nitric acid, by applying a constant current of 40 μA. Regenerated electrodes were immersed in mercury-containing solutions, and were subjected again to electrochemical treatment. This was followed by another regeneration step in pure acid solution, to assess the re-use.

Retrieval tests were also carried out with higher surface area working electrodes employing platinum nanoparticles on carbon black. A 1 mol L$^{-1}$ nitric acid solution containing 10 mg L$^{-1}$ mercury was used as electrolyte.

**Analytical techniques**. The electrolyte aliquots collected were further diluted with high purity 0.5 mol L$^{-1}$ hydrochloric acid, and analysed using Inductively Coupled Plasma-Mass Spectrometry (iCAP Q, Thermo Fischer). The platinum electrodes were investigated before and after electrochemical treatment using X-Ray Diffraction (Siemens Diffraktometer D5000) and Scanning Electron Mycroscopy/Energy Disperssive Spectroscopy (FEI Quanta 200 F/Oxford Inca 300 EDS System).

## Data availability
The authors declare that the main data supporting the findings of this study are available within the article and in the Supplementary Information. Additional information and supporting data are available from the corresponding author ?(B.W.) upon request.

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

## Acknowledgements

The authors gratefully acknowledge financial support from the ÅForsk foundation and Magnus Bergvalls Stiftelse. Adam Arvidsson is acknowledged for his contributions to the illustrations of the atomic structures. Mattias Bengtsson is acknowledged for his contribution to the retrieval study at elevated temperature.

## Author contributions

C.T. and B.W. contributed to the electrode design, and preparation of the electrodes, and designed the experimental procedure. C.T. conducted the electrochemical experiments, and the XRD, SEM/EDS, and ICP-MS analysis. B.W. contributed to the SEM/EDS and electrochemical measurements. Both authors discussed the results, and wrote the manuscript.

## Additional information

**Competing interests:** A patent application (EP 17199244.9) has been filed, and both authors are listed as inventors.

