## [Peer Review File · Nature Communications]

Reviewers' comments:

Reviewer #1 (Remarks to the Author):

I am of opinion favorable to the publication of this manuscript in the e-NATURE without revisions.

I would just like some remarks to be forwarded to the authors and they would send us an answer regarding some points concerning the text.

1- The authors should consider the fact that the acidity of the reaction medium used for the removal of mercury is somewhat high. Therefore, there will be a need to neutralize the acid, in the effective treatment of water, natural, polluted or residual; provided that there is a need for neutralization; still be taken into account the fact that nitrate, in turn, is undesirable for the environment (eutrophication of waters)? On the other hand, some interfering ions and even the Hg (II) ion have a characteristic ionic species distribution as a function of pH, which could hinder the process when the experiments are carried out in neutral or basic pH (Baes, CFJr and Mesmer, RE, The Hydrolysis of Cations, JW & Sons, New York, 1976)

2- With regards the mechanism the text does not seem to make this mechanism so clear, although it points out some bibliographical references that deal with the theme.

My comments:

- With regard to the mechanism proposed for the whole process, in the electrochemical deposition of Hg, it must be observed that the mercury after its reduction to metallic Hg will cover the surface of the substrate, partially or totally, as a film of Hg. After this, the amalgam Pt (Hg) will be formed spontaneously ($\Delta G < 0$), giving rise to the diffusion of Hg into the crystalline lattice of the substrate, solvating the Pt atom and removing the atom to the surface of the substrate to form the species PtHg, PtHg₂ and, or PtHg₄, if there is sufficient bulk Hg to form each species; in excess of Hg the PtHg₄ species is preferably formed. See Arvia et al., they have an important discussion about PtHG intermetallics formation via the voltammetry technique.

- Another important aspect that the authors should take into account in the formation of PtHG intermetallics is the presence of Hg activity covering the surface of the intermetallic film. The presence of this Hg activity species when the film has a long time of permanence on the surface of the substrate of Pt, in the absence of applied potential, promotes the reversal of the reaction decomposing the intermetallic in its original species, Pt and Hg; since Hg diffuses spontaneously to the sublayers of the substrate, in a slow kinetics. I think this fact is important when effectively operating a macroscopic system to remove Hg from large sample volumes; I mean the waiting time between one step and another in the overall industrial process.

- Another aspect to be considered by the authors is that, Lahiri S.K., Gupta D., in his article, 1980, claim to have removed all Hg from the substrate. The presence of Hg in the sublayers distorts the crystalline lattice of Pt and facilitates the attack by nitric acid solution, in the presence of low content of chloride ions, on the substrate by totally removing the Hg from the structure. Thus, if this condition does not apply (HNO₃ / Cl⁻) Hg will not be completely removed either by chemical or electrochemical route; being necessary high temperatures, above 900 °C for its almost total removal.

3- The authors, in my personal opinion, effectively call the attention of the reader to the exclusive formation of the intermetallic PtHg₄. However, the following species will always be present onto/into the substrate of Pt: Hg in the sublayers of the crystalline lattice of Pt; of the amalgam Pt (Hg) and the intermetallic PtHg species. What is occurring in this case is that the thickness of the formed film (750 nm) makes XRD difficult to penetrate and therefore makes it difficult to see what is below the film. On the other hand, the XRD shown in Figure 2, regenerated curve (green curve) clearly reveals the presence of phase reminiscent on the substrate, after removal of PtHg₄, confirming the non-complete electrochemistry removal of Hg. However, this does not invalidate the authors' proposal.

4 - Another aspect to be considered by the authors is the formation of intermetallic Pt_yCu_x (article submitted to Thermochemica Acta) that are formed in the presence of Hg and are not eliminated during the electrochemical removal of Hg. I can suppose that the presence of these intermetallic

species may act in the reuse of the electrode in the next applications. I think that these intermetallics could be act as a diffusion barrier of Hg to the sublayers of the platinum lattice, minimizing the formation of PtHg intermetallics and reducing the efficiency of the electrode to the next applications

5- The authors inform that the electrode can be reused countless times, but do not numerically inform this, so I would like to know how many times the authors could reuse the electrode (s) after the application of samples containing toxic metal ions (Cd (II), Cu (II), etc.)?

Reviewer #2 (Remarks to the Author):

The manuscript describes a study on the potential of removal of mercury (Hg) from aqueous solution via electrochemical alloy formation on platinum.

The authors used platinum electrodes and platinum coated black carbon to remove Hg from water at different concentrations. They found out that the Hg removal process is based on the formation of PtHg₄ which is stable at high temperatures and in a wide pH range. Moreover, they could show that the Hg removal from solution is relatively fast and that the capacity of platinum for Hg binding as an alloy is extremely high. From their results they concluded Pt-Hg-alloy formation is a suitable method to remove Hg from contaminated streams.

The idea of Hg removal from contaminated water through amalgamation is not new. Previous studies used tin, brass or other metal in filters. All these studies have shown that Hg removal through amalgamation is a fast process and that all amalgams generally have a high Hg retention capacity. However, these studies have also shown that amalgams can have several disadvantages such as the dissolution of other metals due to oxidation/Hg reduction, pH dependent dissolution of the amalgams and mechanical instability of the amalgam at high Hg loading.

The amalgamation method suggested by the authors has the advantage that the electrons come from an electrode and no other metal has to be oxidized to reduce Hg²⁺ in solution, so that the amalgam can be formed, which avoids dissolution of other metals. Moreover, different from previous studies Pt-amalgams show high stability in a wide pH range which allows application of the method in a wide pH range. In this sense the suggested method is advantageous compared to previous attempts.

However, there are some open questions regarding the applicability of the method in real world cases. In most cases where Hg removal from contaminated water is necessary large volumes of water have to be processed. To keep dimensions of such filtering systems reasonable the contact time between the filter material and the Hg containing water is usually very short (seconds). The removal times given by the authors e.g. 75 % removal of a 0.05 mg/L Hg solution within one day is relatively long even if platinum nanoparticles on black carbon are used which have been shown to be 20 times faster. Based on these results, I think only the application of platinum nanoparticles on black carbon is suitable for the technical application (waste water treatment or contaminated groundwater). Thus, it would be useful that the authors expand a bit more on the question if large systems of the proposed method, which can process amounts of water in the qm range are possible and if the costs of such systems including regeneration would be reasonable.

Another important point which should be addressed is the application of the method to remove Hg from what the authors called "aqueous streams". What exactly is meant here? The authors mentioned large scale application relevant for decontamination of industrial wastes and drinking water. If this includes natural streams I think the method would not be suitable. In natural waters > 99 % of Hg is tightly bound to dissolved organic complexes which cannot be removed from solution through amalgamation because the Hg cannot be reduced to metallic Hg which is necessary to form the alloy. The same is likely to be valid for industrial wastes and drinking water. The authors should be more specific here and should clearly state for which Hg species the method can be applied.

- What was the divalent Hg species used for the experiments?

- I suggest to cite the following reference:

Zierhut et al. (2009), Activated Gold Surfaces for the Direct Preconcentration of Mercury Species from Natural Waters. *Journal of Analytical Atomic Spectrometry* 24(6)

The manuscript is in principle suitable for nature communications, well written and technically mostly clear.

I suggest possible publication if the raised concerns can be addressed.

Reviewer #3 (Remarks to the Author):

In the ms. the authors report an effective and alternative method to retrieve mercury from water. The technical presentation of the paper is on a good level and it was supported by a considerable amount of experimental data. The subject of the paper is of environmental and societal importance, so is of interest to others in the community. My recommendation is accepted after some minor review. My comments and suggestion are reported hereafter.

Review's comments:

Abstract and Main Text:

1. At my best knowledge, the retrieval of mercury by electrochemical formation of PtHg₄ was document for the first time in 1999 by Fertoni and co-authors. For this reason, the authors should avoid use words like "novel process" in the abstract-line 14 or "a new method" in line 78.
2. Line 95. I found the sentence confused. Please clarify.

Decontamination of different initial concentrations:

3. I found that the time needed it to achieve the equilibrium (except for the initial concentration of 0.05 mg/L) was too long and clearly a limitation to the feasibility of the process. At the end of the ms., the authors highlighted how to solve that issue by evaluating the retrieval of mercury on electrodes with Pt nanoparticles with good results. My question is why the paper is mostly focused on the Pt thin film electrode and not on the electrode with a larger surface area (Pt nanoparticles), the most promising electrode?
4. Please correct the caption of figure 1: (b, bottom left); check the initial concentration of mercury - different values on the plot and on the caption 0.05 or 5 mg/L; (c, right).

Decontamination of different initial concentrations:

5. Why the pH range studied was 0 – 6.6? Why the authors did not consider alkaline values? Mercury precipitation for the highest initial concentrations could be a problem and should be evaluated. Saline waters could have pH values around 8.

Interferences during alloy formation:

6. In the ms., the authors mentioned several times the practical application of this method to retrieve mercury from water. However, the authors only considered cationic ions in the interference study. I think that the presence of anions such as Cl⁻ (that changes Hg speciation) and organic matter (frequently present in natural waters) should be evaluated as well. This will give important information regarding the real feasibility of the proposed method. I believe that the lack of this information is one of the less positive aspects of the work.

Regeneration of used electrodes:

7. Lines 282-283: Which were the good results? Please quantify. And how many cycles retrieve-regeneration were done?

Proof-of-concept:

8. This section is missing. In my opinion, the authors should evaluate the efficiency of the method using real water. This would be "the icing on the cake"

Response to Referees

We would like to thank the Reviewers for their valuable comments. We took all comments into consideration and have revised the manuscript accordingly. To further confirm our claims, we have performed additional investigations, as suggested by the Reviewers. You will find our answers to the reviewer comments and explanations to the modifications done in the manuscript below. We have copied the reviewer comments in black and our response is written in red. The modifications made in the manuscript have been highlighted in the new version.

Reviewer #1

I am of opinion favorable to the publication of this manuscript in the e-NATURE without revisions. I would just like some remarks to be forwarded to the authors and they would send us an answer regarding some points concerning the text.

1- The authors should consider the fact that the acidity of the reaction medium used for the removal of mercury is somewhat high. Therefore, there will be a need to neutralize the acid, in the effective treatment of water, natural, polluted or residual; provided that there is a need for neutralization; still be taken into account the fact that nitrate, in turn, is undesirable for the environment (eutrophication of waters)? On the other hand, some interfering ions and even the Hg (II) ion have a characteristic ionic species distribution as a function of pH, which could hinder the process when the experiments are carried out in neutral or basic pH (Baes, CFJr and Mesmer, RE, The Hydrolysis of Cations, JW & Sons, New York, 1976)

We thank the reviewer for pointing out this important fact. We chose to focus on nitrate-based solutions with pH up to 7 due to good stability of mercury in such media. Some experiments were carried out for long time (one week) and we wanted to make sure that precipitation, absorption, autoreduction and volatilization etc. did not occur during this time. At high pH and or/low concentrations of mercury or in the presence of certain ionic species (e.g., sulfate), aforementioned processes can occur. We wanted to avoid this to get a clear assessment of the alloying process. In the revised version, we have also included a retrieval test from solution containing excess chloride (1 mol/L) at 0.1 mol/L acidity. The excess chloride, which will lead to stronger mercury chloride complexes (compared to nitrates) and mercury chloroanions, did not hinder the process or the overall efficiency over time.

2- With regards the mechanism the text does not seem to make this mechanism so clear, although it points out some bibliographical references that deal with the theme.

My comments:

- With regard to the mechanism proposed for the whole process, in the electrochemical deposition of Hg, it must be observed that the mercury after its reduction to metallic Hg will cover the surface of the substrate, partially or totally, as a film of Hg. After this, the amalgam Pt (Hg) will be formed spontaneously ($\Delta G < 0$), giving rise to the diffusion of Hg into the crystalline lattice of the substrate, solvating the Pt atom and removing the atom to the surface of the substrate to form the species PtHg, PtHg₂ and, or PtHg₄, if there is sufficient bulk Hg to form each species; in excess of Hg the

PtHg₄ species is preferably formed. See Arvia et al., they have an important discussion about PtHg intermetallics formation via the voltammetry technique.

This is true and the bibliographic references in the original manuscript refer to this. In the revised version, we have clarified this further by adding more information and the reference suggested. The concentrations of mercury and the alloying time used in this work were sufficiently high to form PtHg₄, which is thermodynamically favored over the other intermetallic species.

- Another important aspect that the authors should take into account in the formation of PtHg intermetallics is the presence of Hg activity covering the surface of the intermetallic film. The presence of this Hg activity species when the film has a long time of permanence on the surface of the substrate of Pt, in the absence of applied potential, promotes the reversal of the reaction decomposing the intermetallic in its original species, Pt and Hg; since Hg diffuses spontaneously to the sublayers of the substrate, in a slow kinetics. I think this fact is important when effectively operating a macroscopic system to remove Hg from large sample volumes; I mean the waiting time between one step and another in the overall industrial process.

This may be the case in a macroscopic system (and at very low mercury concentrations). To negate this, there needs to be a constant potential to drive the formation of the alloy and prevent the reverse reaction. Given the low currents needed (Fig 5 in the revised manuscript) and assuming the conductivity of the media is adequate throughout the process, this shouldn't pose problems.

- Another aspect to be considered by the authors is that, Lahiri S.K., Gupta D., in his article, 1980, claim to have removed all Hg from the substrate. The presence of Hg in the sublayers distorts the crystalline lattice of Pt and facilitates the attack by nitric acid solution, in the presence of low content of chloride ions, on the substrate by totally removing the Hg from the structure. Thus, if this condition does not apply (HNO₃ / Cl⁻) Hg will not be completely removed either by chemical or electrochemical route; being necessary high temperatures, above 900 °C for its almost total removal.

We agree with the reviewer and our results also indicate that there could be small amounts of Hg left after the regeneration. However, we believe this may not be a problem in real applications. It is fine if, after use, the platinum layer is not 100 % regenerated. This will not hinder the use of electrodes for further decontamination. Since the alloy can be decomposed at elevated temperatures, the platinum/clean electrode can also be reclaimed this way after several electrochemical loading-unloading cycles.

3- The authors, in my personal opinion, effectively call the attention of the reader to the exclusive formation of the intermetallic PtHg₄. However, the following species will always be present onto/into the substrate of Pt: Hg in the sublayers of the crystalline lattice of Pt; of the amalgam Pt (Hg) and the intermetallic PtHg species. What is occurring in this case is that the thickness of the formed film (750 nm) makes XRD difficult to penetrate and therefore makes it difficult to see what is below the film. On the other hand, the XRD shown in Figure 2, regenerated curve (green curve) clearly reveals the presence of phase reminiscent on the substrate, after removal of PtHg₄, confirming the non-complete electrochemistry removal of Hg. However, this does not invalidate the authors' proposal.

We agree with the reviewer on this point and we would like to highlight that we do not claim the exclusive formation of PtHg₄. We do not rule out the formation of other intermetallic species. This paragraph, taken from the original submission, also states this:

'Figure 2 shows the X-ray diffraction (XRD) patterns of a working electrode before and after electrochemical treatment. The predominant platinum-mercury phase formed was PtHg₄, as expected from the thermodynamics (24). While it is possible to form other alloy phases under electrochemical conditions, e.g., PtHg₂, we did not see any clear signs of phases other than PtHg₄. This is likely to be explained by the fact that the experiments were carried out for long times. As PtHg₄ is the alloy phase with the lowest energy, it is expected that this dominates if the system is allowed time to relax, and relatively thick films are formed.'

It is worth mentioning that the XRD in Fig 2 was performed on a film loaded at 22.5 % of the PtHg₄ stoichiometric saturation limit, thus less thickness (750 nm is for 100 % Pt saturated to PtHg₄). This was further clarified in the revised version. The aforementioned text now includes this line: *'The PtHg₄ pattern was registered on a 100 nm platinum electrode after electrochemical treatment, where the platinum film was loaded at about 22.5 % of the PtHg₄ stoichiometric saturation limit during 122 h'*.

4 - Another aspect to be considered by the authors is the formation of intermetallic Pt_yCu_x (article submitted to *Thermochimica Acta*) that are formed in the presence of Hg and are not eliminated during the electrochemical removal of Hg. I can suppose that the presence of these intermetallic species may act in the reuse of the electrode in the next applications. I think that these intermetallics could be act as a diffusion barrier of Hg to the sublayers of the platinum lattice, minimizing the formation of PtHg intermetallics and reducing the efficiency of the electrode to the next applications

That is an interesting observation. We understand that this work is not published yet and we look forward to reading the article. Nonetheless, this will not hinder the use of our process on streams where Cu is not present and streams with low amounts of Cu. As previously mentioned, if the electrochemical release of mercury becomes a problem, the Pt-Hg alloy can be thermally decomposed to reclaim the volatile mercury (as stated in the conclusions section).

5- The authors inform that the electrode can be reused countless times, but do not numerically inform this, so I would like to know how many times the authors could reuse the electrode (s) after the application of samples containing toxic metal ions (Cd (II), Cu (II), etc.)?

We thank the reviewer for pointing this out. In the subsection 'Regeneration of used electrodes' we clarified that, after a loading-unloading cycle, another loading-unloading cycle was performed with comparable efficiency. In the conclusion section, we have indeed mentioned the electrodes can be re-used '*numerous times*'. This statement has been removed from the revised version. The conclusion section contains additional information about re-use, including the thermal recovery of volatile metallic mercury and metallic platinum.

Reviewer #2 (Remarks to the Author):

The manuscript describes a study on the potential of removal of mercury (Hg) from aqueous solution via electrochemical alloy formation on platinum.

The authors used platinum electrodes and platinum coated black carbon to remove Hg from water at different concentrations. They found out that the Hg removal process is based on the formation of PtHg₄ which is stable at high temperatures and in a wide pH range. Moreover, they could show that the Hg removal from solution is relatively fast and that the capacity of platinum for Hg binding as an alloy is extremely high. From their results they concluded Pt-Hg-alloy formation is a suitable method to remove Hg from contaminated streams.

The idea of Hg removal from contaminated water through amalgamation is not new. Previous studies used tin, brass or other metal in filters. All these studies have shown that Hg removal through amalgamation is a fast process and that all amalgams generally have a high Hg retention capacity. However, these studies have also shown that amalgams can have several disadvantages such as the dissolution of other metals due to oxidation/Hg reduction, pH dependent dissolution of the amalgams and mechanical instability of the amalgam at high Hg loading.

The amalgamation method suggested by the authors has the advantage that the electrons come from an electrode and no other metal has to be oxidized to reduce Hg²⁺ in solution, so that the amalgam can be formed, which avoids dissolution of other metals. Moreover, different from previous studies Pt-amalgams show high stability in a wide pH range which allows application of the method in a wide pH range. In this sense the suggested method is advantageous compared to previous attempts.

As Reviewer #2 points out, removal of metals via amalgamation was previously studied. Decontamination studies with tin, brass and gold are most common but no reports on platinum has been presented so far. The *Introduction* section mentions such studies, together with their shortcomings (also pointed out by the Reviewer in his comment). The novelty we want to highlight here is the way amalgamation is done to achieve decontamination. The process we describe is based on potential-controlled electrochemical alloy formation. Decontamination via Pt-Hg alloying under continuous controlled potential has notable advantages, e.g., selectivity and stability. When there is no potential applied, the solubility of mercury in platinum and of platinum in mercury are very low at ambient conditions. By applying a potential to platinum, it is possible to increase the saturation solubility immensely. Moreover, the Pt-Hg system has 12 times higher loading capacity than the Au-Hg system described in some publication. To our knowledge, there are no reports about formation of PtHg₄ under controlled potential for decontamination applications. Another important point is that we show that the process works well also for aqueous streams containing ppm/ppb trace levels of mercury, which are concentrations important from environmental points of view. Of course, studies dealing with the characterization of the species formed at the Pt-Hg interface at given potentials have been published, and we refer to these in our manuscript. However, these were typically done using concentrated monovalent mercury (about 12 g/L) and short contact times (about 300 s). Also, these were not done for, and are not directed towards actual decontamination applications. We mentioned all these aspects in the *Introduction* section.

Please see also below our answer to Reviewer #3. The text adds here include further clarifications to this question.

However, there are some open questions regarding the applicability of the method in real world cases. In most cases where Hg removal from contaminated water is necessary large volumes of water have to be processed. To keep dimensions of such filtering systems reasonable the contact time between the filter material and the Hg containing water is usually very short (seconds). The removal

times given by the authors e.g. 75 % removal of a 0.05 mg/L Hg solution within one day is relatively long even if platinum nanoparticles on black carbon are used which have been shown to be 20 times faster. Based on these results, I think only the application of platinum nanoparticles on black carbon is suitable for the technical application (waste water treatment or contaminated groundwater). Thus, it would be useful that the authors expand a bit more on the question if large systems of the proposed method, which can process amounts of water in the qm range are possible and if the costs of such systems including regeneration would be reasonable.

Indeed, for real applications, a large surface area is needed. We are afraid that, without first developing and tuning a prototype, cost estimations will carry quite large uncertainties. We would like to avoid this in the present manuscript. Here, the main focus is to present the fundamentals of the process and understand how it is affected at the atomic level by various parameters important in real life applications. As written below (please see our answer for Reviewer #3), we wanted to get an understanding of the kinetics, pH dependency, regeneration and re-use, ionic and anionic interferences and draw comparisons with other systems (Au-Hg). Our future goal is to start developing, testing and optimizing a prototype for larger-scale applications. That will be a separate study, in which we will delve deep into applicability.

Another important point which should be addressed is the application of the method to remove Hg from what the authors called "aqueous streams". What exactly is meant here? The authors mentioned large scale application relevant for decontamination of industrial wastes and drinking water. If this includes natural streams I think the method would not be suitable. In natural waters > 99 % of Hg is tightly bound to dissolved organic complexes which cannot be removed from solution through amalgamation because the Hg cannot be reduced to metallic Hg which is necessary to form the alloy. The same is likely to be valid for industrial wastes and drinking water. The authors should be more specific here and should clearly state for which Hg species the method can be applied.

We thank the reviewer for pointing this out and we have clarified some of these aspects in the revised version. Throughout the manuscript, we have re-phrased/further elaborated on the term 'aqueous streams' and 'natural waters' (e.g., please see the changes in the *Conclusions* section). We highlighted the applicability of the method on waters contaminated with inorganic mercury species and natural waters in which mercury is preferably present as inorganic species. We have addressed issues related to mercury speciation in streams containing organic matter (please see the *Interferences* section, as well as our answer to Reviewer #3).

- What was the divalent Hg species used for the experiments?

The divalent mercury compound was mercuric nitrate. This is mentioned in the *Methods/Aqueous feeds* paragraph.

- I suggest to cite the following reference:

Zierhut et al. (2009), Activated Gold Surfaces for the Direct Preconcentration of Mercury Species from Natural Waters. *Journal of Analytical Atomic Spectrometry* 24(6)

We thank the reviewer for this recommendation. We have read the paper and found it very interesting. We have added the reference and a descriptive paragraph.

The manuscript is in principle suitable for nature communications, well written and technically mostly clear.

I suggest possible publication if the raised concerns can be addressed.

Reviewer #3 (Remarks to the Author):

In the ms. the authors report an effective and alternative method to retrieve mercury from water. The technical presentation of the paper is on a good level and it was supported by a considerable amount of experimental data. The subject of the paper is of environmental and societal importance, so is of interest to others in the community. My recommendation is accepted after some minor review. My comments and suggestion are reported hereafter.

Review's comments:

Abstract and Main Text:

1. At my best knowledge, the retrieval of mercury by electrochemical formation of PtHg₄ was document for the first time in 1999 by Fertonani and co-authors. For this reason, the authors should avoid use words like "novel process" in the abstract-line 14 or "a new method" in line 78.

This was partly address above, in our answer for Reviewer #2. We wish to expand the discussion here. Indeed, electrochemical formation of platinum-mercury alloys were reported by Fertonani and his group in the 90s. Platinum-mercury solid-state interactions were of interest in past years with the aim to obtain stable, uniform, and thin mercury films on platinum for analytical studies. This is important for achieving higher hydrogen overpotential, allowing voltammetric studies at more cathodic potentials, and anodic stripping analysis of more active metals. The works of Fertonani et al. and Souza et al. (all cited in the *Introduction* section) focused on characterizing the compounds that form at the platinum-mercury interface using XRD, SEM, XPS and TGA. These studies used very concentrated monovalent mercury solutions (12 g/L) and an electrochemical potential applied for short times (300 s). While the authors confirm formation of PtHg₄ and other intermetallic species, they do not focus on potential applications for decontamination of aqueous streams; nor do they conduct further studies relevant to this (influence of pH, uptake time, presence of other ionic species in solution, electrochemical regeneration and re-use of the platinum substrate, influence of mercury concentration in solution, uptake from solutions containing trace levels of mercury, complete saturation of the substrate etc.). We thoroughly address all these aspects in our manuscript. As mentioned, we did not find any papers showing the potential-controlled Pt-Hg system for decontamination of solutions containing amounts of mercury relevant from an environmental perspective (ppb-ppm). For these reasons we consider the application of the method to be '*novel*'. We have added these clarifications in the *Introduction*, before Equation 1.

We have modified the formulation in the abstract from '*novel process*' to '*novel application*' to clarify that the novelty lies in the actual application of the electrochemical amalgamation process. We have deleted the word 'new' from line 78. In light of these discussions, if the Editors and the Reviewers find the word '*novel*' to still be out of place, we can remove it.

2. Line 95. I found the sentence confused. Please clarify.

This sentence reports how many atoms of platinum are soluble in liquid mercury. We have rephrased this to make this clearer.

Decontamination of different initial concentrations:

3. I found that the time needed it to achieve the equilibrium (except for the initial concentration of 0.05 mg/L) was too long and clearly a limitation to the feasibility of the process. At the end of the ms., the authors highlighted how to solve that issue by evaluating the retrieval of mercury on electrodes with Pt nanoparticles with good results. My question is why the paper is mostly focused on the Pt thin film electrode and not on the electrode with a larger surface area (Pt nanoparticles), the most promising electrode?

The focus of this paper was to study the fundamentals of the decontamination process and to see what happens at molecular level. We wanted to get an understanding of the kinetics, pH dependency, regeneration and re-use, cationic/anionic interferences etc. For such studies, a well-defined and well-controlled flat surface is preferred, even as the relatively low surface area leads to long reaction times. We believe these aspects are very important for understanding the system and its real-life applicability. Once we had these answers, as proof-of-concept, we did a successful decontamination trial with electrodes with large surface area.

4. Please correct the caption of figure 1: (b, bottom left); check the initial concentration of mercury - different values on the plot and on the caption 0.05 or 5 mg/L; (c, right).

This has been corrected.

Decontamination of different initial concentrations:

5. Why the pH range studied was 0 – 6.6? Why the authors did not consider alkaline values? Mercury precipitation for the highest initial concentrations could be a problem and should be evaluated. Saline waters could have pH values around 8.

We chose to work in nitrate media and in this pH range for stability reasons, e.g., to avoid precipitation/absorption of mercury and make a better assessment of the electrochemical amalgamation process. For the trace level experiments (50 ppb mercury in solution), the pH was kept low (pH 0) to avoid absorption on the walls of the electrochemical cell. The more concentrated solutions (e.g., 20 ppm) were also prepared in 1 mol/L nitric acid. At such low pH and concentrations, precipitation and absorption of mercury is not an issue during the investigated time. At higher pH, we kept a high ionic strength, 1 mol/L nitrate. We did not notice any precipitation losses and the levels of mercury were sufficiently high (10 ppm) to minimize errors from potential absorption of ppt/few ppb amounts of mercury. We wanted to have a good comparison between the systems (similar ionic strength and mercury content) so we chose to not increase the pH higher due to the risk of precipitation and unwanted absorption.

Interferences during alloy formation:

6. In the ms., the authors mentioned several times the practical application of this method to retrieve mercury from water. However, the authors only considered cationic ions in the interference study. I think that the presence of anions such as Cl⁻ (that changes Hg speciation) and organic matter (frequently present in natural waters) should be evaluated as well. This will give important

information regarding the real feasibility of the proposed method. I believe that the lack of this information is one of the less positive aspects of the work.

We agree with the reviewer and thank for this correct observation. In the revised manuscript we have included a test at high chloride concentration (0.1 mol/L HNO₃, 1 mol/L NaCl). This was done to draw comparisons with the 0.1 mol/L HNO₃, 1 mol/L nitrate system from the original manuscript, and to see how the changes in mercury speciation affect the process. This is described in detail in the *Interferences during alloy formation* section. To accommodate the new results, Fig. 3 was split into the two consisting parts (a and b); the new test was grouped with Fig 3b into Fig 4. As reported, the process was not hindered by the high chloride concentration and formation of anionic mercury species, e.g., (HgCl₄)²⁻.

We were not able to investigate a system where mercury is bound to organic matter. We agree with the observations made by the Reviewer, and we addressed this in the manuscript. We do not expect the alloying process to completely stop but the formation rate may differ (depending on how easily organomercury ions (CH₃Hg⁺) are attracted to the working electrode and broken under potential). If the organic mercury ions are at the surface of the electrode/can easily reach the surface, and the decomposition energetics are favorable, alloy formation should be possible. Zierhut et al. (Activated gold surfaces for the direct preconcentration of mercury species from natural waters, 2009; work cited and discussed in the revised version of the manuscript) found that '*The percentage of adsorption and adsorption mechanism of different mercury species onto gold is dependent on the morphology of the gold surface. Smooth surfaces are selective towards elemental mercury, whereas nano-structured gold surfaces retain different mercury species (Hg⁰, Hg²⁺, MeHg⁺) with comparable adsorption rates. Nano-structured gold surfaces are catalytically active with regard to stripping the methyl group from MeHg⁺ and reducing mercury species to Hg⁰.*' Albeit directed to Au-Pt, this finding is important. We already noticed several parallels between the Pt-Hg system and the Au-Hg system reported by others (please see the original manuscript, e.g., lines 159-172). It is worth mentioning that organic mercury compounds are photodegradable under UV light. This can be exploited to assure the process proposed here can still find applications for such streams. The following discussion was added in the revised version:

'Natural waters typically contain organic matter which facilitates formation of organic mercury species, e.g., methylmercury (CH₃Hg⁺). How the presence of such species in solution affects the alloying process was not studied here. Mercury in aquatic environments cycles between various chemical species; photodecomposition of methylmercury has been reported (37, 38). Decomposition of methylmercurychloride to divalent mercury can take place immediately under UV exposure, depending on the intensity of the radiation (37). An in-situ study of lake water showed that the annual rates of methylmercury photodegradation in surface waters are almost double the estimated external inputs of methylmercury from rain, snow, streamflow and land runoff (38). For these reasons, we do not completely rule out the applicability of the method on contaminated solutions containing organic matter. Mercury levels in such streams can still be decreased under UV irradiation, as the ionic inorganic species from the photodegradation of organic mercury will form platinum alloys. If the organic species can easily move to the surface of the working electrode (e.g., attraction of the positive methylmercury ion), and the energetics favor decomposition, alloy formation should also occur in the absence of UV radiation.'

Regeneration of used electrodes:

7. Lines 282-283: Which were the good results? Please quantify. And how many cycles retrieve-regeneration were done?

This was clarified in the text. After a loading-unloading cycle, another loading-unloading cycle was performed with comparable overall efficiencies.

Proof-of-concept:

8. This section is missing. In my opinion, the authors should evaluate the efficiency of the method using real water. This would be “the icing on the cake”

We agree with the reviewer that such an experiment would be very interesting to perform. Unfortunately, we have not been able to find a suitable stream of real water to evaluate the process on.

REVIEWERS' COMMENTS:

Reviewer #1 (Remarks to the Author):

Dear authors, thank you for your attention and your kindness in the response.

All of this evaluator's suggestions were met and in perfect compliance.

I congratulate the authors for the quality of their work.

Reviewer #3 (Remarks to the Author):

I was very pleased with the replies given by the authors.

I have no more comments on the revised manuscript and I recommend its publication.

Responses to Referees

The authors would like to thank the referees again for their valuable comments and suggestions. After the previous revision, no further issues were raised by the referees.